# Obesity and Energy Substrate Transporters in Ovarian Cancer—Review

**DOI:** 10.3390/molecules26061659

**Published:** 2021-03-16

**Authors:** Marta Baczewska, Klaudia Bojczuk, Adrian Kołakowski, Jakub Dobroch, Paweł Guzik, Paweł Knapp

**Affiliations:** 1Department of Gynecology and Gynecological Oncology, Medical University of Białystok, 15-089 Bialystok, Poland; klaudia.bojczuk98@wp.pl (K.B.); adriankolakowski17@gmail.com (A.K.); jakub.dobroch@umb.edu.pl (J.D.); knapp@umb.edu.pl (P.K.); 2Clinical Department of Gynecology and Obstetrics, City Hospital, 35-241 Rzeszów, Poland; pawelguzik@gmail.com; 3University Oncology Center, University Clinical Hospital in Białystok, 15-276 Białystok, Poland

**Keywords:** ovarian cancer, obesity, cancer progression, targeted therapy, lipids

## Abstract

Ovarian cancer is the seventh most common cancer in women. It is characterized by a high mortality rate because of its aggressiveness and advanced stage at the time of diagnosis. It is a nonhomogenous group of neoplasms and, of which the molecular basics are still being investigated. Nowadays, the golden standard in the treatment is debulking cytoreductive surgery combined with platinum-based chemotherapy. We have presented the interactions and the resulting perspectives between fatty acid transporters, glucose transporters and ovarian cancer cells. Studies have shown the association between a lipid-rich environment and cancer progression, which suggests the use of correspondent transporter inhibitors as promising chemotherapeutic agents. This review summarizes preclinical and clinical studies highlighting the role of fatty acid transport proteins and glucose transporters in development, growth, metastasizing and its potential use in targeted therapies of ovarian cancer.

## 1. Introduction

Ovarian cancer (OC) is the seventh most common cancer in women and the most common cause of death from gynecological cancers, with a 5-year survival rate below 45% [1,2]. 90% of ovarian cancers are epithelial cancers, the most common, of which are high-grade serous ovarian cancer (HGSOC), known among healthcare professionals as the silent killer [3]. According to US data from 2016 covering the entire female population in the US for every 100,000 women, 11 new ovarian cancer cases were reported, and 7 died of cancer. Late diagnosis of advanced disease is the main cause of poor prognosis. In the early stages of the disease, patients do not present any symptoms; the screening test does not exist. Over the last two decades, ovarian cancer rates have decreased in North America and Europe [4]. Approximately half of the epithelial ovarian cancers (EOC) had defects in DNA repairing systems, while 96% of HGSOC tumors have TP53 mutation and present impaired apoptosis [5,6]. The recent studies focus on the investigation of the metabolic basis of OC. Observation of diverse clinical conditions in patients with an equal histopathological status suggests the potential difference between preferred energy substrates. Factors as hypoxia, oxidative stress or inflammation generally redirect the cell metabolism into anaerobic processes and enhance the role of glucose. However, the bioavailability of free fatty acids increases analogically in a neoplastic environment. The aim of this work is to present the current scientific approach to OC metabolism and the potential clinical application.

## 2. From Diagnosis to Setting a Proper Treatment Plan in OC

### 2.1. Signs and Symptoms

Although ovarian cancer can occur in younger women, it has a predisposition to develop in women over 50 and with menopause, which means that as life expectancy increases, the number of cases diagnosed increases each year. The vast majority of ovarian cancers are diagnosed at an advanced stage due to the asymptomatic course of the early-stage and nonspecific symptoms of the advanced stage [3,7]. The presence of symptoms, such as pelvic pain, constipation, diarrhea, nausea, urinary problems, early satiety, suggest performing a physical examination and transvaginal ultrasounds initially [8,9].

### 2.2. Diagnosis

Thereafter, ovarian cancer diagnosis includes contrast computed tomography (CT) or magnetic resonance imaging (MRI) of the chest, abdomen, and pelvis to determine cancer stage and the presence of metastases. It is also advisable to test tumor markers, such as CA-125, CEA and CA 19–9, in order to exclude other causes of abdominal mass symptoms different from ovarian cancer. Additionally, the severity of the disease and potential resectability should be elucidated prior to laparotomy. Laparoscopic evaluation of the abdominal cavity or percutaneous biopsy in the case of disseminated disease should be considered. Examination of abdominal fluid samples in patients with ascites can also be used for cachectic patients but is less reliable [10]. It is vital to provide genetic counseling for women with a family history associated with a risk of harmful BRCA1/2 mutations and HR deficiency (HRD), which increases the likelihood of developing ovarian cancer [11].

### 2.3. Surgical Treatment and Chemotherapy

Nowadays, the golden standard in the treatment is debulking cytoreductive surgery combined with platinum-based chemotherapy (cisplatin, carboplatin or oxaliplatin) and a taxane in 4–8 cycles. Over the last two decades, guidelines have changed significantly [12]. In advanced stages of OC, neoadjuvant therapy may be required before the surgery in order to reduce tumor mass [8]. Cancer recurrence and platinum resistance are common burdens in women diagnosed in the advanced stage. The scientific findings in the area of molecular and genetic alternations in OC suggest that the potential beneficial effect of targeted therapies should be inevitably evaluated [13].

Vasculogenesis and angiogenesis mediated by vascular epithelial growth factor (VEGF) has a prominent meaning in the epithelial ovarian cancer development and spread. The tumor blood vessels are more prone to VEGF effects than normal ones. Inhibition of vascular epithelial growth factor receptors (VEGFR), highly expressed in OC, decreases tumor vessels or metastases formation and cancer progression. Monoclonal antibody bevacizumab inhibits tumor VEGFR [11]. AURELIA trial reveals that adding bevacizumab to conventional chemotherapy in platinum-resistant OC enhances response to the treatment and increases progression-free survival [14].

### 2.4. Maintenance Treatment

Novel and constantly evolving treatments for primary and recurrent ovarian cancer include angiogenesis inhibitors, poly (ADP-ribose) polymerase (PARP) inhibitors and immunotherapy agents [6,15]. Furthermore, PARP inhibitors, such as olaparib, niraparib and rucaparib, play an essential modulatory role by inhibiting DNA repairing systems [1]. Since BRCA mutations lead to cells’ DNA double-strand breaks, PARP inhibitors can cause the death of those cells by leaving their DNA damaged [13]. A current state of knowledge shows that PARP inhibitors have an application in the treatment of patients with BRCA1/2 mutations and recurrent ovarian cancer [1]. There is a strong need to elucidate novel therapies based on inhibiting cell proliferation and angiogenesis because of their potential application.

## 3. Obesity and Ovarian Cancer

Obesity poses a threat for diverse tumor development and is correlated with dismal prognosis [16]. The high concentration of adipocytes in the human organism results in adipose tissue impairment, which leads to immune and hormonal alternations in the microenvironment that is a fundamental part of carcinogenesis [16]. However, the excessive visceral fat distribution is suspected to increase the likelihood of cancer development, but a general higher fat concentration in the whole human body is not correlated with this risk [17]. Besides the widespread conviction that the OC is an obesity-related neoplasm, the meta-analyses do not confirm this theory. Nevertheless, the association between increased body weight and ovarian cancer exists in the premenopausal period. High-grade invasive serous tumors, the most fatal subtype, in this study were not associated with BMI [18]. Bae et al. also showed that BMI at diagnosis could not be a prognostic factor for the survival of ovarian cancer patients [19]. However, there are numerous studies describing an increased risk of ovarian cancer in obese women. Foong et al. demonstrated obesity as a potentially modifiable malignant factor in ovarian cancer in their meta-analysis [20]. Moreover, the literature reviews show conflicting reports on correlations between obesity and ovarian cancer, so more research is required to confirm this thesis. It is worth mentioning that the main methods of obesity measurement in the above-mentioned works were BMI and WHR. It is crucial that those indexes do not provide data on fat distribution and may vary depending on the clinical condition. In women suspicious of OC, the WHR is not adequate either due to ascites or mass in the abdomen. Dual-energy X-ray absorptiometry seems to be an adequate method to evaluate the exact amount of visceral fat but is hardly accessible. It is a high-quality alternative to the reference methods of measuring visceral adipose tissue, which is CT and MRI [21,22]. There are studies suggesting that future research should be more precise, unified and include features as the histological type and menopausal status [23].

The recent studies focus on the investigation of the metabolic basis of OC. The observation of diverse clinical conditions in patients with an equal histopathological status suggests the potential difference between preferred energy substrates. It is commonly known that cancer cells have increased requirements for ATP production and higher energy supply. Warburg et al. observed that malignant cell transformation can lead to higher glucose utilization via glycolysis even under normoxia [24]. Furthermore, other diverse coexistent factors involved in tumor development like hypoxia, oxidative stress or inflammation also redirect the cell metabolism into anaerobic processes and enhance the role of glucose [25,26]. Since cancerous cells derive energy mostly from glycolysis, they have a predilection to elevated glucose utilization. From the observations mentioned above, it can be proved that the cancer cells alter the expression of glucose transporters in order to increase glucose influx [27]. It can be observed that the overexpression of different facilitative glucose transporters (GLUT) has been discovered in diverse cancer tissues. Glucose transporter 1 (GLUT 1) is characteristic of liver, pancreas, kidney, lung and glucose transporter 3 (GLUT 3) of lung and ovarian neoplasms. Unfortunately, the higher presence of GLUTs in cancer tissues is not precisely evaluated, and further research should be conducted to provide more detailed data [28]. However, the bioavailability of free fatty acids (FFA) increases analogically in a neoplastic environment. Visceral adipose tissue surrounding tumors has a changed phenotype induced by cancerous cells. Cancer-associated adipocytes produce FFA and adipokines [26]. FFA can be a source of energy for cancerous tissues by fatty acid oxidation but also occupy a central part in a cell structure synthesis and mediate by fatty acid-binding protein (FABP) signaling pathways involved in cancer development [29]. Moreover, overexpression of fatty acid-binding protein 4 (FABP4) was observed in different cancers, e.g., prostate, bladder, renal cell carcinoma and also ovarian neoplasms. FABP4 is suspected to be a potential point of alternative treatments for cancers associated with the adipose microenvironment [30].

## 4. Glucose Metabolism in Cancer Cells

It is widely known that malignant transformation can lead to increased metabolism [27]. Furthermore, malignant cells are characteristic of higher glucose consumption via anaerobic processes, such as glycolysis even under normoxia, which is known as the Warburg effect [31]. Anaerobic glucose consumption results in less ATP production than oxidative phosphorylation [32]. Moreover, diverse environmental factors, such as hypoxia, inflammation, stress and lack of nutrients, also shift cancer metabolism into anaerobic. As a result of the mentioned observations, it is known that cancer cells require a huge amount of glucose [25,33,34]. Glucose influx into the cell is the first limiting process in the glycolysis pathway in non-malignant and cancerous tissues mediated by facilitative glucose transporters across the plasma membrane [27,35]. In the current state of knowledge, there are 14 GLUTs in mammalian cells involved in glucose transport, which have diverse functions in glucose uptake and also vary in their location in mammalian tissues, regulation and their affinities for glucose [36,37]. There are a few GLUTs overexpressed in cancer cells involved in glucose uptake, especially GLUT 1–4, in order to elevate glucose uptake [36,38]. Moreover, alternations in GLUTs expression in cancers are caused by diverse pathways. Influence of hypoxic environment on cancerous cells results in the inhibition of hypoxia-induced factor (HIF) alfa and beta degradation, which in this condition bind to hypoxia–response elements (HRE) and induce transcription of glycolytic genes like GLUT 1 and 3 [27,39,40]. PI3K and Akt pathway and p53 mutation also contribute to enhanced transcription of GLUT 1 and 3 [40]. The translocation of glucose transporter 4 (GLUT 4) to the plasma membrane in insulin-dependent tissues is related to insulin stimulation; however, factors, which induce the expression of GLUT 4 in cancerous tissues should be assessed [40]. Taking into account that malignant cells are highly dependent on glycolysis as the main way of energy production and that many factors regulate glucose uptake, there is the likelihood that the inhibition of proteins involved in glycolysis, such as glucose transporters, can have beneficial effects in cancer therapy [33].

### 4.1. A Broad Role of GLUT 1 in Cancer Development

GLUT 1 is abundantly found in the brain and erythrocytes, but also in adipocytes, liver and muscles and plays a vital role in basal glucose influx into the cell [27,41]. Furthermore, GLUT 1 is also regarded as a prerequisite in diverse processes of cancer development [42]. First of all, GLUT 1 is involved in enhanced glucose influx in malignant cells induced by genetic alternations, growth factors and also hypoxia. GLUT 1 also occupies a central part in energy production necessary for the proliferation of cancer cells [28,40]. Nogushi et al. proved that the presence of antisense GLUT 1 in mice resulted in the reduction of tumor growth with concomitant alternations in the cell cycle and lower glucose influx observed in a gastric carcinoma cell line with antisense GLUT 1. These observations confirm that the expression of GLUT 1 is a fundamental part of processes involved in tumor development [43]. Moreover, Tsukioka et al. revealed that there is a remarkable correlation between GLUT 1 expression and VEGF in EOC. GLUT 1 is possibly involved in new vessel formation in EOS via VEGF, which is a potent factor of neovascularization in EOC [42,44,45]. Additionally, GLUT 1 is also suspected to be a crucial factor in metastasis formation. Ito et al. detected that enhanced expression of GLUT 1 in rhabdomyosarcoma cell line was coupled with an elevated level of MMP-2 protein targeting degradation of collagen in the extracellular matrix and suspected to regulate cancer cell migration and invasion [46,47,48]. All of the observations above confirm that glucose uptake mediated by GLUT 1 plays a crucial role in cancer development, especially EOC, as well as in metastasis formation.

### 4.2. GLUT 1 in OC

The most prominent issue worth mentioning is that OC significantly varies from other types of cancers. OC is more prone to forming metastases by the intraperitoneal spread. Taking this into account, it can be supposed that OC cells and intraperitoneal metastases are more deprived of oxygen supply if they are located far away from blood vessels. Kalir et al. showed that expression of GLUT 1 was observed in 96% of the human OC, lower expression of GLUT 1 was detected in a borderline tumor, and local expression of GLUT 1 was also observed in villus adenomas with the hazard of malignant transformation [49]. However, expression of GLUT 1 was observed only in 57% of specimens from the human esophageal squamous cell carcinomas in 60% of the human gastric cancer samples [24,50]. Higher expression of GLUT 1 in OC induced by hypoxia is thought to be an adaptive mechanism to oxygen deprivation observed in OC [49]. Furthermore, other research also indicates that that GLUT 1 is highly upregulated in malignant OC [51]. Cantuaria et al. revealed in their study the presence of GLUT 1 in 101 out of 103 specimens of the human EOC. Their findings also proved significant alternations in GLUT-1 expression in malignant rather than borderline and benign tumors [35]. In other studies, the expression of GLUT 1 was observed in 98.7% of the human EOC [44]. A significant increase in expression of GLUT 1 was also observed in well-differentiated OC [35]. There is also a vast majority of studies that show that there is a correlation between overexpression of GLUT 1 and the histological type of OC. GLUT1 is abundantly expressed in serous adenocarcinoma; however, expression of these transporters is decreased in clear cell adenocarcinoma [35,42,44,51,52]. There is also a correlation between overexpression of GLUT 1 and stage of OC. GLUT 1 is upregulated in an advanced stage of OC rather than in the early stages, but that observation was only detected in serous adenocarcinomas [33,44,51,53].

GLUT 1 is regarded as a prognostic factor and associated with dismal prognosis in some cancers, e.g., lung cancer, colon cancer, gastric cancer, renal [52,54,55,56]. Yin et al. revealed that the positive effect of the treatment was lower in gastric cancer with expression with GLUT 1 [55]. Some data indicate there is a possible connection between overexpression of GLUT 1 and prognosis in OC; however, that statement is unclear [51,53]. Cho et al. showed that overexpression of GLUT 1 in EOS is a prognostic factor of unfavorable prognosis, but assessing this marker has some analytic burdens [51]. Semaan et al. showed that advanced stages of OC with concomitant overexpression of GLUT 1 are less prone to chemotherapy, but one marker is not an adequate predictor in EOC [42]. Xintaropoulou et al. proved that there is no remarkable correlation between the expression of GLUT 1 in OC and patient survival [33]. Tsusioka showed that the evaluation of GLUT 1 expression in OC is inadequate for predicting prognosis because other factors like age, stage of cancer and histological type have a far more prognostic value for survival in OC [44].

Summing up, OC is highly dependent on glucose consumption, and overexpression of GLUT 1 is more obvious than in other cancers. Some types of OC, such as serous adenocarcinoma and advanced stages of OC, have a predilection for overexpressing GLUT 1. The application of novel glycolytic inhibitors as an additive therapy to conventional chemotherapy may have the clinical application [33,57,58]. However, the expression of GLUT 1 is not regarded as an adequate predictor of patient survival in OC. Understanding the fact that OC is highly reliant on glycolysis as a source of energy and that GLUT 1 plays a significant role in cancer development, some researchers checked whether the inhibition of GLUT 1 results in alternations in cancer cell proliferation and suppression of tumor growth. Some studies indicate that the inhibition of GLUT 1 has a potential clinical application in OC treatment if there is a significant overexpression of GLUT 1 in OC. Shin et al. proved that ciglitazone, an antidiabetic drug, can also play a crucial role in the inhibition of tumor proliferation in vitro by altering glucose uptake mediated by GLUT 1, but also by changing the amount of GLUT 1 in the plasma membrane. Furthermore, it was also observed that the application of ciglitazone in mice in vitro resulted in OC size reduction [57]. Ma et al. revealed that the application of a GLUT 1 inhibitor BAY-876 in OC in both in vitro and in vivo models in mice resulted in a 50–71% decrease in tumor growth and also reduction of glucose utilization rate [58]. Other research revealed that administration of STF 31, novel GLUT 1 inhibitor and metformin to conventional chemotherapy caused the amelioration of therapy effects in both platinum-sensitive and resistant OC [33]. All of the information above confirms that the inhibition of glucose uptake mediated by GLUT 1 plays a significant role in the suppression of energy production in OC. The administration of GLUT 1 inhibitors can reduce cell proliferation and cancer growth. Resistance to platinum poses a threat in conventional OC chemotherapy, and the fact that platinum-resistant OC is still prone to glycolytic pathway inhibitors can cause beneficial development of conventional treatment in OC.

### 4.3. A Potential Role of GLUT 3 in OC and Other Cancers

The numerous research indicates an inevitable role of GLUT 1 in cancer development, whereas a precise function of other facilitative glucose transporters in that process is still not well understood [34]. GLUT 3 is also abundantly found in tissues with higher requirements for glucose, e.g., the brain, because of its higher affinity for glucose and plays a crucial role in basal glucose influx [41,59]. The presence of GLUT 3 was also detected in diverse cancers, e.g., lung cancer, endometrial cancer, and gastric cancer and was associated with dismal prognosis [34,41,44]. GLUT 3 not only plays a crucial role in glucose uptake in cancer cells but also, similarly to GLUT 1, has a remarkable correlation with the level of VEGF, which is regarded as a vital factor of cancer neovascularization. Schlößer et al. revealed the presence of GLUT 3 in 66% of the human primary gastric cancer or adenocarcinoma of the gastroesophageal junction, and higher expression of GLUT 3 was associated with advanced UICC stage of cancer. Furthermore, the expression of GLUT 3 in primary gastric cancer has a remarkable correlation with unfavorable patients’ survival [34]. However, studies conducted on the human stage I non-small-cell lung carcinoma revealed that the presence of GLUT 3 in only 21% of specimens [41]. According to Tsukioka et al., GLUT 3 is also expressed in OC, and the presence of GLUT 3 was revealed in 92.8% of the human EOC specimens in their study. However, Rudlowski et al. proved the homogeneous, weak expression of GLUT 3 in the human malignant OC, and interestingly, the presence of this glucose transporter does not differ from benign lesions [53]. The current research does not prove any correlations between the expression of GLUT 3 and cancer stage or histological type. Taking into account unclear data about the expression of GLUT 3 in cancers, there is a strong need to conduct further research to confirm a precise function of GLUT 3 and whether targeting therapy by GLUT 3 inhibition may have potential clinical application in OC treatment.

### 4.4. A Role of GLUT 4 in OC and Other Cancers

GLUT 4 is facilitative glucose transporter predominantly expressed in insulin-dependent organs, such as brown and white adipose, skeletal and cardiac muscle [60]. GLUT 4 is found primarily in a tubulo-vesicular compartment in cells. Translocation of this transporter to the plasma membrane is mediated by insulin stimulation, which results in enhanced glucose uptake to stimulated cells [40,60]. The presence of GLUT 4 is also abundantly found in some cancerous tissues, which interestingly do not respond to insulin effects in physiological conditions, e.g., colon, lymphoid, breast, thyroid, pancreatic and gastric carcinoma [28,40,61,62,63,64]. The current state of knowledge indicates the presence and a crucial function of GLUT 4 in OC; however, precise data about these glucose transporters in carcinogenesis have been little studied, and data are still unclear [44,65]. Tsukioka et al. in revealed a higher presence of GLUT 4 in 84.4% of the human EOC specimens. Moreover, GLUT 4, similarly to GLUT 1, has a remarkable correlation with the level of VEGF, which is responsible for angiogenesis in cancer development and metastases formation [44]. This fact is confirmed by the observation that suppression of VEGFR2/AKT1/GSK3β/SOX5/GLUT 4 pathway results in attenuating tumor growth in OC [66]. Furthermore, expression of GLUT 4 varies in different histological types of OC and is significantly higher in advanced stages of OC. However, Rudlowski et al. revealed that GLUT 4 is not expressed in malignant OC [53]. There are also some studies that confirm an increased expression of GLUT 4 in malignant OC than in benign and borderline. However, in this study, the presence of GLUT 4 was higher in mucinous and clear cell adenocarcinomas, which are associated with inferior response to chemotherapy. Whereas GLUT 1 was detected to be mostly expressed in serous adenocarcinoma. These studies indicated that there is the likelihood that overexpression of GLUT 1 and 4 is triggered by different factors in OC development [44,65]. Novel research should be conducted in order to confirm a precise function of enhanced expression of GLUT4 in OC development and provide information on whether inhibition of GLUT 4 will have beneficial effects in transporter-targeted therapies in some types of OC associated with resistance to chemotherapy because some research shows overexpression of GLUT 4 in mucinous and clear cell adenocarcinomas [65].

Understanding these facts in OC metabolism, some researchers conducted research to prove whether inhibition of GLUT 4 results in alternations in cancer cell proliferation and suppression of tumor growth. Chen et al. revealed that the impact of apatinib on OC, a tyrosine kinase inhibitor, resulted in a decrease in glucose consumption and attenuation of cancer cell proliferation in vivo in OC cells and also tumor growth suppression in vitro in mice. Apatinib was regarded in this study as a modulatory factor in glucose metabolism in OC by suppressing VEGFR2/AKT1/GSK3β/SOX5/GLUT4 pathway [66].

The vital role of glucose and its transporters—GLUTs in OC metabolism is inevitable. The current state of knowledge shows that overexpression and the role of GLUT1 are most obvious in OC, and a precise function of other facilitative glucose transporters in OC development should be assessed [34]. Resistance to chemotherapy poses a hazardous problem in OC treatment, so there is a strong need to elucidate whether a therapy targeting glycolysis pathway could have a clinical application because there are some studies that confirm its potential, beneficial effect [33,57,58,66] in Table 1. 

## Potential Inhibitors

### 4.5. A Role of MCT in OC and Other Cancers

Most cancerous cells derive energy mainly from glycolysis and have an elevated glucose utilization rate [27]. As a result of glycolysis due to the Warburg effect in cancerous cells (metabolism of glucose to lactic acid even under normoxia), a huge amount of lactic acid is produced in the malignant cells [67]. Too much lactic acid may be detrimental to cells by changing pH in intracellular fluid; thus, there is a need to efflux excessive lactate to tumor environment, vascular endothelial cells, and to an oxidative phenotype of cancer cells via monocarboxylate transporters (MCTs), mainly MCT1 and MCT 4 [68,69]. Some studies show that there is a significant overexpression of MCT1, MCT2, and MCT4 in some cancers, e.g., MCT4 in adrenocortical cancer, MCT1 and MCT 4 in cervix cancer, MCT1 and MCT 2 in brain cancer as adaptation strategy [70,71,72,73]. Monocarboxylate transporters play a predominant role in regulating pH balance in cancer cells in the case of enhanced glucose glycolysis and lactate production [67]. Moreover, the current state of knowledge indicates that lactate acts as a modulatory factor in cancerogenesis. Hence, lactate induces cytokines and growth factors secretion by macrophages, which promotes inflammation development in the tumor environment, tumor cell growth and metastasis formation [74]. Furthermore, Pinheiro et al. discovered a significant correlation between CD147 and MCT1 expressions in ovarian cancer. Interestingly, overexpressed CD147 induces the production of metalloproteinases and VEGF in cancerous tissues, and therefore, it induces cancer development and aggressiveness [67]. All of the observations mentioned above confirm that lactate metabolism contributes to cancer metastases formation. Hence, MCT’s relevance in cancer development should be thoroughly elucidated.

## 5. Adipocytes and the Role of Lipid Transporters

Interestingly, the adipocytes play a key role in OC metastasis to the omentum (Figure 1). The experiments in mice showed that injection of OC cells in association with omental adipocytes resulted in three times bigger size of the tumor than OC alone [75]. This outcome suggests the role of tumor lipid metabolism is dependent not only on genetic and epigenetic changes but also on the bioavailability of lipids. The existence of adipose tissue in the environment of the tumor in light of current knowledge is inevitable. Taking into account the collective data describing an interaction between ovarian tumor and adipose tissue, further studies of biology and ovarian cancer cell metabolism should be conducted.

### 5.1. Fatty Acid Binding Protein 4

Despite a lack of strong evidence on the correlation between body mass and HGSC, metastatic tissues are characterized by increased expression of FABP4. The experiments on mice proved the role of FABP4 in the metastatic potential of cancer cells [76]. FABP4 is an adipocyte isoform of fatty acid-binding protein (A-FABP/FABP4/aP2). The function of FABP4 is to promote the uptake of long-chain fatty acids and to participate in lipid transport and metabolic regulation. Much research was done recently to explain the biochemical pathways and significance of this protein in cancerogenesis. There is inconsistent data about the role of FABP4 in ovarian cancer cells. Yu et al. indicate that the increased level of this protein in ovarian cancer cells is a result of inflammation, more specifically IL17-A activity and was associated with FAs intake followed by cell growth [77]. On the contrary, Hua et al. showed that the inhibition of SRC (proto-oncogene protein tyrosine kinase Src) was associated with increased levels of FABP4 in non-small cell and renal cell carcinoma. They suggest that intracellular FABP4 plays a key role in decreasing lipid droplets, which is accompanied by the formation of ROS (reactive oxygen species) [78] (Figure 2). It is widely known that the rapid proliferation of cancer cells is associated with increased levels of ROS. To survive, the cell must activate rescue metabolic pathways to reduce the amount of ROS. The safe pathways for OC as the Warburg effect (glucose to lactate transformation observed in cancer cells) and aerobic glycolysis are induced by SRC. Therefore, treatment with SCR-inhibitor resulted in decreased tumor growth in vivo. Moreover, simultaneously treating OC with FABP4-inhibitor and SCR-inhibitor showed additive growth suppression, the opposite effect they expected. The confirmation of the inhibitory effect of FABP4 suppression on the growth of ovarian cancer cells is the work described by Mukherjee et al. [79]. Targeted therapy on FABP4 was performed in cancer cells grown together with primary human omental adipocytes. This resulted in increased levels of 5-hydroxymethylcytosine in DNA and decreased the number of clone formation and gene signatures associated with ovarian cancer metastasis. This resulted in a reduction in tumor size and metastatic potential. It is worth mentioning that the FABP4 inhibitor significantly increased the sensitivity of cancer cells to carboplatin. Taken together, these studies suggest that the adaptation of ovarian cancer cells to a lipid-rich environment is accompanied by an increased concentration of FABP4. Inhibition of these lipid transporter proteins reduces the aggressiveness of ovarian cancer, its metastasis to the peritoneum and other high-fat environments. Finally, it reduces the size of the tumor. It is worth mentioning, according to Nieman et al., we do not observe an increased concentration of FABP4 in ovarian cancer cells alone, neither in those accompanying adipocyte-depleted tissues [75]. This observation indicates the key role of FABP4 in peritoneal metastases and a large association with obesity, where there is a significant increase in the number of adipocytes.

### 5.2. CD36

The CD 36 receptor is an 88 kDa membrane glycoprotein and is a scavenger receptor that binds to various ligands, such as apoptotic cells, thrombospondin-1 (TSP-1) and FFA. CD36 is expressed in multiple cell types, mediates the binding and cellular uptake of long-chain fatty acids, oxidized lipids and phospholipids, advanced oxidation protein products and has roles in lipid accumulation, inflammation, apoptosis and molecular adhesion [80]. CD36 mediates the absorption of fatty acids, the major nutrients for the tumor. The fatty acids are derived from tumor-associated adipocytes and, in high concentrations, promote the proliferation and metastasis of tumor cells. This fatty acids transporter is also important in the presence of tumors by accelerating tumor growth but also inhibits angiogenesis and promotes vascular apoptosis when TSP-1 binds to CD36 on the surface of the MVEC [81]. Pascual et al. described a high ability to initiate metastasis at high concentrations of CD36, which can either be caused by a high-fat diet or induced by palmitic acid in mouse models of human oral cancer. It has also been shown that the use of CD36 inhibitors virtually completely inhibits metastasis [82]. Ladanyi et al. reported in their work that OC cells co-cultured with primary human network adipocytes showed high concentrations of CD36 in the cell membrane, which increased fatty acid uptake. The use of CD36 inhibitors prevented the development of the malignant phenotype by reducing the accumulation of lipid droplets but also reducing the concentration of ROS [83]. Last, but not least, CD36 shows a higher concentration and is more prominent in peritoneal metastases of ovarian cancer than in primary ovarian cancer and normal tissue [84]. To sum up, CD36 is another fatty acid transporter whose increased concentration is observed in the presence of adipocytes. This leads to the development of a malignant type of tumor and metastases, most often to the peritoneum. According to observations mentioned above, CD36-targeted therapy is one of the methods of treating cancer as it can both reduce the size of a tumor and prevent or inhibit the development of a malignant tumor.

### 5.3. Fatty Acid-Binding Protein 6 (FABP 6)

FABP 6 is found in adipocytes but less expressed than FABP4 and macrophages. FABP 6 is produced in the liver, then released with bile to the small gut and is involved in micelles formation. FABP 6 mediates bile acid transport in an ileal epithelium. It is commonly known that bile acids are regarded as a crucial factor initiating inflammation in colonic epithelial cells and cell death [85]. As a result of inflammation-induced in colonic cells, bile acids cause oxidative DNA damage, which is a fundamental part of malignant transformation [86,87]. Overexpression of FABP 6 is suspected to be generated by the influence of bile acids on colon cancer cells in animal models [88]. According to Ohmachi et al., FABP 6 is abundantly found in colon cancer cells compared with noncancerous tissues. Their study also revealed that there is a remarkable correlation between the size of the tumor and the expression of FABP6; more precisely, smaller tumors have an enhanced expression of FABP 6. Interestingly, levels of FABP6 expression significantly differ depending on the histological type of colorectal cancer. The current research does not prove any correlations between the expression of FABP 6 and cancer stage [87]. It is commonly known that FABP6 is present in the cell, but no less important is its elevated blood level during cancer progression. These observations confirm the recent research, which indicated FABP4 and FABP6 as potential independent biomarkers of colorectal cancer. The increased levels in the serum were associated with a higher risk of this neoplasm. Interestingly, the levels of those proteins after the surgery were significantly reduced [89].

However, there is no research that indicates any positive correlation between elevated levels of FABP6 and malignant transformation in OC. There is a strong need to conduct further research to confirm the precise function of FABP 6 in cancer development and whether there is any correlation between ovarian cancer and enhanced concentration of FABP 6 in the serum.

### 5.4. Role of FFA Oxidation in Ovarian Cancer Progression

The high mortality rate of ovarian cancer results from the late detection of the disease when it is in a highly advanced stage and metastasized. The spread of ovarian cancer in the body depends on the survival of the epithelial cells after their detachment and loss of nutrients due to cellular stress. Ovarian cancer cells become resistant to apoptosis as a result of fatty acid oxidation by increasing the concentration of one of the essential beta-oxidation enzymes [90]. An example of such an enzyme is carnitine palmitoyltransferase 1 (CPT1), which catalyzes the transfer of the long-chain acyl group of an acyl-CoA ester to carnitine. Studies have shown increased levels of this enzyme in OC cell lines as well as decreased survival in patients with overexpression of this enzyme. It follows that CPT1 may be a potential marker and a potential target of ovarian cancer therapy in order to limit the spread of neoplastic cells [91]. Feng et al. showed in their work that an increase in the concentration of CPT1 and Acyl-CoA dehydrogenase enzymes (ACAD enzymes) promotes epithelial ovarian cancer progression [84]. However, the function of all the enzymes involved in beta-oxidation is not fully elucidated, and data are not clear. Zhang et al. described that a low concentration of carnitine palmitoyltransferase 2 (CPT2) correlates with a decreased survival of patients with ovarian cancer and an increased frequency of metastases. This is because CPT2 suppresses the G1/S cell cycle transition as well as induces cell apoptosis. According to the authors of this study, CPT2 has the opposite effect on CPT1 and ACAD enzymes and inhibits tumor growth and metastasis dissemination [92]. It is worth mentioning that some enzymes and molecules that induce beta-oxidation can cause resistance to some anticancer drugs. A great example is collagen type XI alpha 1 (COL11A1), which is a new biomarker of cisplatin resistance in ovarian cancer. COL11A1 is both an inducer of beta fatty acid oxidation and increases the expression of proteins involved in the synthesis of fatty acids. COL11A1-induced resistance to cisplatin can be abolished by inhibiting beta fatty acid oxidation [93].

The modification of fatty acid oxidation for the therapeutic target in ovarian cancer patients has not yet been fully investigated and requires further research. The effect of individual enzymes on ovarian cancer cells varies, and the concentration of these enzymes must be increased or decreased depending on the enzyme. As a result, they seem to be a good therapeutic direction in targeted therapy and in the treatment of resistance to certain anticancer drugs.

## 6. Role of Adipokines

Adipose tissue is not only involved in energy storage but also plays a crucial role in hormone production, which are called adipokines [94]. Among many of them, leptin and adiponectin have the most diverse and well-known effects on our bodies [95]. Adiponectin is responsible, among others, for inducing insulin-sensitivity, antiinflammatory reactions and regulating energy metabolism [96,97]. It was shown that reduced adiponectin levels and high visceral fat could induce insulin resistance and β-cell failure [98]. There is a significant decrease in adiponectin concentrations in the serum associated with obesity [95]. The current state of knowledge indicates that adiponectins also have antitumor properties in obesity-associated cancers, such as breast, endometrial cancer, by inhibiting the ERK1/2-MAPK pathway [99]. Recent studies also indicate that adiponectins are also involved in the regulation of cancer cell invasion by inhibiting vascular endothelial growth factor, which is a potent factor initiating neovascularization [100]. Hoffman et al. proved that adiponectin suppresses ovarian cancer cell proliferation by decreasing the expression of receptors for insulin-like growth factor and estradiol [99]. Jin et al. revealed a significant decrease in adiponectin concentrations in the serum in the ovarian cancer patients compared with the control group. Moreover, their study does not prove any remarkable correlation between the stage of ovarian cancer and adiponectin level in the serum [95]. However, Tiwari et al. did not reveal any alternations in adiponectin levels in the serum in chicken models [101]. Adiponectin induces its effects on tissues by interacting with one of the receptors—adiponectin receptor 1 (AdipoR1) and adiponectin receptor 2 (AdipoR2). There is some research that proves a significant decrease in expression of AdipoR1 and AdipoR2 observed in ovarian cancer cell lines and, more precisely, expression of those receptors was lower in the epithelial ovarian cancer cell line compared with granulosa tumor cell line [99].

Methylation of the DNA is a process that occurs in normal and cancerous cells. The result is a change in the gene function and its final products without a change in the DNA sequence. DNA methyltransferase (DNMT1) is an enzyme that takes part in the methylation of adiponectin, the hormone that has a beneficial effect on the human body. Understanding these facts, the researchers checked whether the epigenetic targeting drugs as guadecitabin (DNMT-Inhibitor) could affect the interaction between adipocytes and OC cells. They showed that methylation is associated indirectly with metastatic cell behavior—increases cell migration and invasion, and that upregulation of suppression gene SUSD2 (Sushi Domain Containing 2) can reduce cancer cell expansion. Guadecitabin possibly changes the intracellular signaling pathways, activates the inflammatory response and indirectly prevents metastasis. These data suggest its potential use in the early-stages of Ovarian Cancer FIGO I/II, especially that authors did not observe its beneficial effect in vitro in cotreatment with alkylating Carboplatin [102].

Leptin is another type of adipokine, which also plays a broad role in normal cells and also acts as a growth factor in cancerous cells. It is involved in energy homeostasis, regulates food intake, but also exerts influence on hematopoiesis, angiogenesis and reproductive system, e.g., regulates the secretion of gonadotropin hormones [95]. The significant alternations in leptin concentration in the serum are suspected to be associated with obesity [103]. What is more, some research indicates that there is also a significant correlation between enhanced levels of leptin and developing cancer associated with obesity [95,104]. Moreover, the secretion of leptin is triggered by insulin, TNF alfa, reproductive hormones, but also by hypoxia via HIF-1, which all are involved in cancer development [104]. Leptin exerts its influence on cells by interacting with one of its receptors—LEPR and impacts diverse pathways, which promote cancerogenesis through activation of the phosphatidylinositol 3-kinase (PI3K) and mitogen-activated protein kinase (MAPK) [105,106]. Choi et al. revealed the expression of leptin receptors in OC and, what is more, that leptin enhances cell proliferation via the ERK1/2 pathway in OC and inhibits cell apoptosis by the inhibition of constitutive phosphorylation of p38 MAPK [107]. Furthermore, leptin is also suspected of inducing the overexpression of MMP-7 protein via ERK and JNK pathways in OC. MMP-7 induces degradation of collagen in the extracellular matrix, and as a result of that, it promotes cancer cell migration and metastasis formation in OC development [108,109]. This finding correlates with some research that proved that the inhibition of leptin-induced pathways suppresses the intraperitoneal spread of OC cells in vivo xenograft. This study revealed that PI3K/Akt/mTOR pathway induced by leptin is involved in OC peritoneal metastasis formation and interestingly showed a novel potential therapeutic target for ovarian cancer [110]. It is worth mentioning that some studies confirm that the incubation of leptin with OC cells treated with paclitaxel decreased the amount of OC cells in the G2/M phase. Paclitaxel exerts its impact on OC cells by slowing down OC cell proliferation rate by blocking the microtubule. Hence, the incubation of leptin with OC cells ameliorates the influence of paclitaxel/Taxol on cancer cell microtubules. It is worth mentioning that high levels of leptin hypothetically may have linkage with resistance to chemotherapy, but these data are unclear and need further research [111].

In summary, there is a vast majority of studies that confirm a remarkable correlation between lower expression of adiponectin, higher expression of leptin and ovarian cancer. However, there is a strong need to conduct further research to confirm the precise function of adiponectin and leptin in ovarian cancer development or whether there is any correlation between the stage of ovarian cancer, histological type and adiponectin, leptin expression. Moreover, it is also crucial to evaluate whether therapy targeting adiponectin and leptin expression will have clinical application in OC treatment.

## 7. Conclusions

Ovarian Cancer is a nonhomogenous disease, and its pathogenesis is still being explored. Among different biochemical phenomena, the research of the lipid and glucose pathways seems to be worth focusing on. Since the link between the expression of fatty acid and glucose transporters in the development and progression of ovarian cancer is widely investigated, many original works proved the role of adipose tissue in tumor growth and metastasis. Obesity and, more precisely, biochemical effects of the coexistence of adipocytes and ovarian cancer cells might be the key to understanding the pathogenesis of neoplastic disease progression. It is indisputable that obesity and its consequences, such as inflammation and ROS production, promote oncogenesis. Nevertheless, due to the rarity of ovarian cancer disease and ambiguous results, well-designed multicenter studies should be carried out to evaluate the precise OC risk factors.

There is a need to conduct further research to investigate new anticancer agents and assess the effects of targeted therapy on fatty acid transporters and glucose transporters. Some results of previous studies were promising because the use of particular inhibitors resulted in a decline in tumor size and a significant reduction in metastasis, mainly to the peritoneum.

Moreover, it is worth pointing out that some of FABPs could be used as potential biomarkers; however, this aspect in OC patients has not been proven yet.

## Figures and Tables

**Figure 1 molecules-26-01659-f001:**
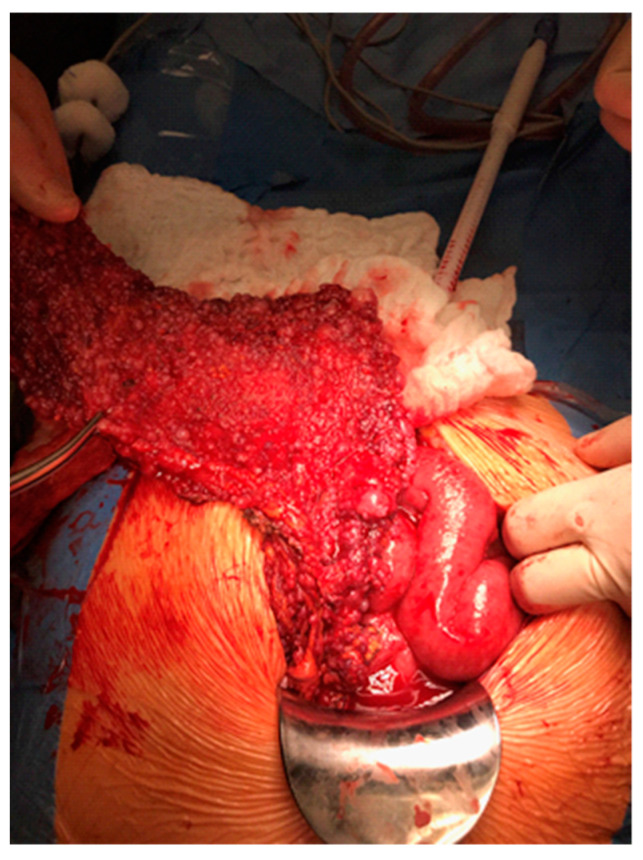
High-grade ovarian cancer‚ omental cake-like metastasis to the omentum in a 68-year old female; FIGO IIIC; 2019. University Oncology Center, Bialystok, Poland.

**Figure 2 molecules-26-01659-f002:**
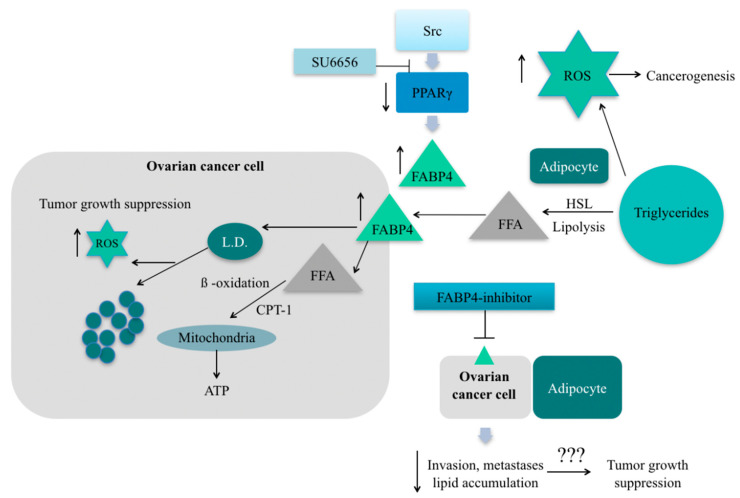
Proposed model of FABP4 role in the ovarian cancer cell [75,78] by Baczewska et al. SU6656—SRC-inhibitor; L.D—lipid droplets; FFA—free fatty acids; HSL—hormone-sensitive lipase.

**Table 1 molecules-26-01659-t001:** Facilitative glucose transporters in ovarian cancer—an overview.

Type of GLUT	Percent of OC with Overexpressed GLUT	Major Subtype of OC with Higher Expression of GLUT	Proposed Role in Dif-ferent Cancer (Not Only in OC)	Potential Inhibitors
GLUT 1	96% [49] 98.7% [44]	- Serous adenocarcinoma [35,42,44,51,52] - Advanced stages of OC (detected only in serous adenocarcinoma) [33,44,51,53]	- Adaptation to hypoxia [49] - Basal influx of glucose, which is a source of energy [28,40] - Possibly involved in mechanism—metastasis formation [46,47,48] - Possibly involved in new vessel formation via VEGF [44]	- Ciglitazone—changing amount of GLUT1 in plasma membrane [57] - BAY-876—reduction of glucose utilization rate [58] - STF31 and metformin—inhibition GLUT1
GLUT 3	92.8% [44]	The current research does not prove any correlations	- Glucose reuptake [44] - Factor of neovascularization via VEGF [44]	not detected
GLUT 4	84.4% [44]	Mucinous and clear cells adenocarcinoma	Angiogenesis, metastasis formation via VEGFR2/AKT1/GSK3β/SOX5/GLUT4 pathway [66]	Apatinib–modulatory factor in VEGFR2/AKT1/GSK3β/SOX5/GLUT4 pathway [66]

## Data Availability

The data presented in this study are available on request from the corresponding author.

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
