# Peer review of "Obesity and Energy Substrate Transporters in Ovarian Cancer—Review"

_molecules, 2021, doi:10.3390/molecules26061659_

Round 1
Reviewer 1 Report
The review by Baczewska et al. discusses obesity and energy substrate transporters in ovarian cancer. This is a pertinent topic to discuss as the crosstalk between adipose/lipid and ovarian cancer is now well established, while on the other hand the relationship of BMI/adiposity and ovarian cancer is not that well established yet. Combined with importance of metabolic adaptations undertaken by cancer cells, including ovarian cancer; the concept of using energy substrate transporters as potential therapeutic targets warrants discussion. While the content of the review is informational, however the quality of writing and paucity of references cited lowers the enthusiasm significantly.
- Many statements made about ovarian cancer do not hold strong. The authors mention ovarian cancer to be “one of the most common neoplasms in women”, which it is not. It is the most lethal of gynecological cancers. This is a cross misinformation, please rectify and provide more recent references. Another example is stating that “ovarian can occur at any age” or “ovarian cancer is a no-heterogenous disease”. Please do review of ovarian cancer correct all such statements.
- The authors have given the major weightage to GLUTs. In the same pathway lactate transporters (MCTs) have been reported to be upregulated in cancer cells and implicated in ovarian and other cancer. Thus, a section on these should be included.
- The level and expression of adiponectin is generally considered along with levels of leptin. Thus, few lines on leptin should be added to the section of adipokines.
- The quality of figures need improvement in terms of fonts/colors etc.
- The manuscript requires major proof-reading. There are hanging sentences, periods where not required, typos and major grammar corrections.
Author Response
Response to Reviewer 1 Comments
Dear Reviewer,
Thank you for giving us the opportunity to submit a revised draft of my manuscript titled Obesity and energy substrate transporters in ovarian cancer - review to Molecules. We appreciate the time and effort that you and the reviewers have dedicated to providing your valuable feedback on my manuscript. We are grateful to the reviewers for their insightful comments on my paper. We have been able to incorporate changes to reflect most of the suggestions provided by the reviewers. We have highlighted the changes within the manuscript using track changes in Microsoft Word. All page numbers refer to the revised manuscript file with tracked changes. We have put the responses to your comments below:
Point 1: Many statements made about ovarian cancer do not hold strong. The authors mention ovarian cancer to be “one of the most common neoplasms in women”, which it is not. It is the most lethal of gynecological cancers. This is a cross misinformation, please rectify and provide more recent references. Another example is stating that “ovarian can occur at any age” or “ovarian cancer is a no-heterogenous disease”. Please do review of ovarian cancer correct all such statements.
Response 1: Thank you for pointing this out. We agree with this comment that we have inserted many statements about ovarian cancer that do not hold strong and were too much generalised, so we have corrected our statements according to recent references. The “one of the most common neoplasms in women” has been corrected to “the seventh most common cancer in women and the most common cause of death from gynecological cancers, with a 5-year survival rate below 45%” according to reference number 1 and 2, lines 26-31.
The “ovarian can occur at any age” has been corrected to “ovarian cancer can occur in younger women, it has a predisposition to develop in women over 50 and with menopause, which means that as life expectancy increases, the number of cases diagnosed increases each year” according to reference number 3 and 7, lines 47-49.Our intention was to show that at an earlier age the incidence of ovarian cancer is lower and increases with each passing year after the menopause.
The “ovarian cancer is a no-heterogenous disease” has been corrected to “Ovarian Cancer is non-homogenous”, line 560. There was probably an auto-correction of the text, because our idea from the beginning was non-homogenous.
Point 2: The authors have given the major weightage to GLUTs. In the same pathway lactate transporters (MCTs) have been reported to be upregulated in cancer cells and implicated in ovarian and other cancer. Thus, a section on these should be included.
Response 2: Thank you for pointing this out. We agree with this comment that we have given the major weightage to GLUTs, whereas the impact of MCTs in cancer development is relevant as well. Therefore, we have created a new section on MCTs influence in ovarian and other cancers and we have inserted it on pages number 8 and 9, paragraph 4.5., lines 326-347.
Point 3: The level and expression of adiponectin is generally considered along with levels of leptin. Thus, few lines on leptin should be added to the section of adipokines.
Response 3: As suggested by the reviewer, we have added description of leptin influence in OC development to the section of adipokines on pages numer 13 and 14, in paragraph 6, lines 523-551.
Point 4: The quality of figures need improvement in terms of fonts/colors etc.
Response 4: Thank you for pointing this out. As suggested by the reviewer, we have updated our figures, improved the colours, font sizes and corrected typing mistakes in Figure 1 on page 2 and Figure 3 on page 11.
Point 5: The manuscript requires major proof-reading. There are hanging sentences, periods where not required, typos and major grammar corrections.
Response 5: : We agree with the reviewer’s assessment. Accordingly, throughout the manuscript, we have revised hanging sentences, typing mistakes and major grammar corrections.
Thank you for your tips for our work. We have tried to make all the corrections as stated in the review. In case of any corrections, we are open to further cooperation.
Please see attachment.

Reviewer 2 Report
In this review manuscript by Baczewska et al., authors summarize preclinical and clinical studies highlighting the role of fatty acid and glucose transporters in development, growth, and metastasis as potential targeted therapies of ovarian cancer.
The review is well written, logical and the figures are helpful. However, the following items need to be considered to improve the manuscript.
The description of all the GLUT genes involved in glucose metabolism is great. However, it would be most helpful to readers in a tabulated form, where the relevance of each GLUT is highlighted.
In the introduction, the authors say “However, the bioavailability of free fatty acids increases analogically in a neoplastic environment.” Lipid metabolism and the FFA availability is indeed important to ovarian cancer progression. However, this review does not consider the utilization of FFA by the cancer cells. Mainly the role of fatty acid oxidation ( CPT1, CPT2, ACAD enzymes) in the progression of the disease. This is a critical component in ovarian cancer with recent papers providing scientific evidence, including a role for fat oxidation in in Taxol therapy resistance. Authors need to consider this addition to the review.
Reference 40 regarding GLUT4 and insulin: the reference (PMID: 17611657) does not indicate that insulin stimulates GLUT4 upregulation in ovarian cancer subtypes. Please clarify.
Author Response
Dear Reviewer,
Thank you for giving us the opportunity to submit a revised draft of my manuscript titled ‘Obesity and energy substrate transporters in ovarian cancer – review’ to Molecules. We appreciate the time and effort that you and the reviewers have dedicated to providing your valuable feedback on my manuscript. We are grateful to the reviewers for their insightful comments on my paper. We have been able to incorporate changes to reflect most of the suggestions provided by the reviewers. We have highlighted the changes within the manuscript using track changes. All page numbers refer to the revised manuscript file with tracked changes. We have put the responses to your comments below:
Point 1: The description of all the GLUT genes involved in glucose metabolism is great. However, it would be most helpful to readers in a tabulated form, where the relevance of each GLUT is highlighted.
Response 1: We agree with this comment that we have updated our manuscript with an overview of GLUTs impact on OC development and we have inserted it on page 8 (Table 1), line 323.
Point 2: In the introduction, the authors say “However, the bioavailability of free fatty acids increases analogically in a neoplastic environment.” Lipid metabolism and the FFA availability is indeed important to ovarian cancer progression. However, this review does not consider the utilization of FFA by the cancer cells. Mainly the role of fatty acid oxidation
(CPT1, CPT2, ACAD enzymes) in the progression of the disease. This is a critical component in ovarian cancer with recent papers providing scientific evidence, including a role for fat oxidation in in Taxol therapy resistance. Authors need to consider this addition to the review.
Response 2: Thanks for paying attention. The reviewer is right and we agree that the oxidation of FFA has a significant influence on the development of ovarian cancer. That's why we created a new section called "Role of FFA oxidation in ovarian cancer progression" for the latest information on the effects of FFA oxidation on ovarian cancer, and put it on page 12 and 13. Unfortunately, we were unable to find information on the effect on taxol resistance. However, we have described the induction of cisplatin resistance by COL11A1.
We have insterted it in lines 454-485.
Point 3: Reference 40 regarding GLUT4 and insulin: the reference (PMID: 17611657) does not indicate that insulin stimulates GLUT4 upregulation in ovarian cancer subtypes. Please clarify.
Response 3: We agree with this comment that reference 40 (PMID: 17611657) does not indicate that insulin stimulates GLUT4 upregulation in ovarian cancer subtypes. The reviewer is correct, first of all our statement ‘overexpression of glucose transporter 4 (GLUT 4) in this case is triggered by insulin is not accurate, we meant that translocation of GLUT 4 to the plasma membrane is stimulated by insulin, however it only takes place in insulin dependent tissues. Our statement was generalised too much because there is a need to conduct research which will discover factors that induce overexpression of GLUT 4 in OC. This statement has been corrected on page 5 in paragraph 4 according to recent reference which in revised manuscript is reference number 40 (PMID: 26773935) (reference numbering changed after adding new points). The correction is put in lines 167-169.
Barron, C. C., Bilan, P. J., Tsakiridis, T., & Tsiani, E. (2016). Facilitative glucose transporters: Implications for cancer detection, prognosis and treatment. Metabolism: clinical and experimental, 65(2), 124–139. https://doi.org/10.1016/j.metabol.2015.10.007
Thank you for your tips for our work. We have tried to make all the corrections as stated in the review. In case of any corrections, we are open to further cooperation.
